# Factors Influenced the Endoscopic Services Volume during the COVID-19 Pandemic at National Tertiary Referral Hospital in Indonesia: Dr. Cipto Mangunkusumo Hospital

**DOI:** 10.3390/healthcare10112280

**Published:** 2022-11-14

**Authors:** Chyntia Olivia Maurine Jasirwan, Amal C. Sjaaf, Anhari Achadi, Prastuti Soewondo, Roswin Rosnim Djaafar, Rino A. Gani

**Affiliations:** 1Hepatobiliary Division, Staff Medic Group of Internal Medicine, Faculty of Medicine, Universitas Indonesia, Dr. Cipto Mangunkusumo Hospital, Jakarta 10430, Indonesia; 2Non-Surgical Interventional Service Installation, Dr. Cipto Mangunkusumo Hospital, Jakarta 10430, Indonesia; 3Hospital Administration Studies, Department of Health Policy Administration, Faculty of Public Health, Universitas Indonesia, Depok 16424, Indonesia; 4Metropolitan Medical Center Hospital, Jakarta 12940, Indonesia

**Keywords:** COVID-19, endoscopic volume, service flow

## Abstract

The impact of the COVID-19 pandemic caused a decrease in healthcare services, the intervention of non-surgical procedures, and endoscopy. This study examined the volume of endoscopy at Dr. Cipto Mangukusumo Hospital, the highest referral hospital in Indonesia. A cross-sectional mixed method was used to assess the relationship between endoscopy volume, age, gender, number of COVID-19 cases, type of patient’s case, the origin of treatment, and the kind of endoscopic procedure before and during the pandemic. The secondary data were collected through the hospital’s Electronic Health Record (EHR) System and “Kawal COVID-19” Websites, while the primary data were collected through observation, document reviews, and in-depth online interviews with doctors at endoscopic units. This study period was divided into six intervals of three months, respectively, from January 2020 to September 2021, and 5030 endoscopic procedures were collected. The data were analyzed both quantitatively through the SPSS statistics and qualitatively. The quantitative data presented as descriptive and bivariate results in an Independent T-Test and a Chi-Square test. The results showed there was a significant difference (*p* = 0.004) in the volume of endoscopes before (the highest volume) and during the pandemic (the lowest volume during April–June 2020 period). The mean age of the patients was higher before the pandemic. There was a significant difference between patient admissions from outpatient and emergency procedures before and during the pandemic. There are changes in the flow of outpatient to do endoscopies which were different from the flow of emergency patients during the pandemic, which focused on the long waiting list for inward entry queues, the mandatory COVID-19 PCR swab, and the criteria of emergency cases for fast-track procedures, the reduced bed capacity, and the expired date of laboratory examinations. The decreased volume was also caused by the limitation of patient intervention by the doctors. However, the duration of the action procedure was accelerated without reducing its quality. Furthermore, there was a high wave of Delta Variant cases from May to July 2021. In addition, the factors of age, type of patient’s case, origin, and treatment showed significant differences before and during the COVID-19 pandemic. Finally, changes in the flow of services also influenced various impacts on endoscopy and service costs. Therefore, further study is required to calculate the unit costs.

## 1. Introduction

In January 2020, a new virus called Severe Acute Respiratory Syndrome Coronavirus-2 (SARS-CoV-2) was identified in Wuhan, Hubei Province, China, and became the cause of the global pneumonia outbreak [1]. The SARS-CoV-2 virus is transmitted through droplets and close contact through activities that produce aerosols, environmental pollution, fomite, and fecal–oral transmission [2].

Since the COVID-19 pandemic began, due to the high risk of infection, many hospitals issued new policies related to the flow of treatment, such as changes in patient transportation in and out, restrictions on the number of people in the procedure room, unfavorable pressure rooms, personal protective equipment, changes in room layout, as well as anesthesia protocols and intubation. In addition, international endoscopic societies have issued various guidelines and recommendations [3,4,5] to reduce the risk of transmission and control COVID-19 infection.

The impact of the COVID-19 pandemic caused a decrease in the number of outpatient visits by 60% around early April 2020, and there was also a sharp decline in hospital revenues [6]. The volume of the endoscopic procedure is one of the performance indicators of the endoscopic procedure unit. The decrease in the volume of endoscopic procedures will be in line with the decrease in income where endoscopic procedures are one of the most profitable non-surgical procedures for hospitals, so an in-depth analysis is needed to examine the magnitude of the problem of decreasing the volume of endoscopy and what factors influence it during the COVID-19 pandemic. Furthermore, it negatively affected gastroenterology services for face-to-face consultations with patients and elective procedures [7]. A study in Italy on 847 patients from 13 endoscopy centers in 2020 showed a decrease in the volume of endoscopic procedures that paralleled the increase in COVID-19 cases due to the selection of patients for endoscopy based on priority (emergency, urgent and elective) [8]. According to Garbey et al. (2020), significant changes in endoscopy services during the COVID-19 pandemic were related to prolonged operating room turnover due to a longer disinfection process. Furthermore, the number of health workers in non-COVID-19 services decreased due to infection, and those experienced stress [9]. The psychological factors and patient discomfort while coming to the hospital also delayed the treatment [10]. Zein et al. (2020) reviewed the preparation and response of endoscopy service units in tertiary hospitals relating to facilities, referrals for complicated cases, and financing for National Health Insurance during the COVID-19 pandemic in Indonesia. However, there has not been a single study on endoscopy services’ outcomes in service volume and performance, revenue, and patient satisfaction.

Based on a preliminary study of monthly endoscopic procedures from the Non-Surgical Intervention Service Installation (NSISSI), which oversees the Integrated Procedure Room (IPR) and the Gastrointestinal Endoscopy Center (GEC) at Dr. Cipto Mangunkusumo Hospital, it was found that there was a 64% drastic decrease in the mean total volume between January to May 2020. Therefore, an in-depth analysis is needed to examine the magnitude of the problem of decreasing endoscopic volume and the factors that influenced it during the pandemic. Therefore, this study further examined changes in endoscopic services to identify service volume patterns and influence factors quantitatively and qualitatively at Dr. Cipto Mangunkusumo Hospital during the COVID-19 pandemic between 2020 and 2021.

## 2. Materials and Methods

### 2.1. Study Design

This is a cross-sectional observational and analytical case study using mixed quantitative and qualitative methods to examine the endoscopic volume pattern throughout 2020 to the end of September 2021. Informed consent was obtained from all subjects involved in the study.

### 2.2. Location and Time

This study was conducted in November 2021 at the Non-Surgical Interventional Service Installation (NSISI), which oversees the Hepatobiliary Procedure Room and the Gastrointestinal Endoscopy Center (GEC) of Dr. Cipto Mangunkusumo Hospital, Jakarta, Indonesia.

The pandemic period is divided into six parts: before the pandemic, January–March 2020 (period A) and after it had occurred between April–June 2020 (period B), July–September 2020 (period C), October–December 2020 (period D), January–March 2021 (period E), April–June 2021 (period F), and July–September 2021 (period G).

### 2.3. Population and Samples

This study includes patients who experienced diagnostic and therapeutic gastroscopy (Esofagogastroduodenoscopy/EGD) procedures (ligation, histoacryl injection, other types of therapeutic endoscopy), diagnostic or therapeutic colonoscopy, Endoscopic Retrograde Cholangio–Pancreatography (ERCP), and Endoscopic Ultrasonography (EUS) from January 2020 to September 2021 in the Integrated Procedure Room of Hepatobiliary and the Gastrointestinal Endoscopy Center (GEC) of Dr. Cipto Mangunkusumo Hospital. The sample was taken using the saturated technique from January 2020 to September 2021. This study did not classify patients undergoing endoscopic procedures as positive or negative for COVID-19. The volume of endoscopic was divided into periods A g with three months intervals from January 2020 to September 2021. Chai et al. report significant differences in gastrointestinal endoscopic procedures between 2019 and 2020 in January–April. Hence, we selected three months as a time interval by considering the previous studies used as a reference for sample calculations. A further reason for selecting the three months interval was the load cases of COVID-19 in Jakarta that was spreading fast, and hospitals’ response in preparing for adaptation regulations has changed quickly. The minimal sample size for the volume of endoscopic services before the pandemic (January–March 2020) and during the pandemic (April 2020–September 2021) was calculated by unpaired numerical comparative test formula resulting in a total number of minimal 48 patients in each period A-G. This study was conducted in one center (Cipto Mangunkusumo Hospital), while a reference study for sample calculation was obtained from 271 multicenter hospitals. Of the data study reference, endoscopic procedures decreased significantly from 3,203,594 (before the pandemic) to 1,512,619 (during the pandemic) in 271 hospitals. Therefore, we performed a minimal sample calculation, but all existing patients who underwent endoscopic procedures in that period were used as overall study samples.

### 2.4. Data Collection Techniques

Data collection techniques were carried out on a secondary and primary basis. The secondary data were collected on the volume of endoscopy services and patient characteristics, types of patient cases, treatment onset, and endoscopic procedures. This was carried out by tracing and retrieving patient data in the hospital’s Electronic Health Record (EHR) System. Furthermore, quantitative data collection relating to the number of COVID-19 cases in Jakarta was obtained through the open access report on the “Kawal” website. Primary data for qualitative analysis about the service flow and implementation of standard operating procedures before and during the COVID-19 pandemic was carried out by observation, review documents, and interviews with nurses in the Hepatobiliary Procedure Room, head of NSISSI, head of administration and finance.

### 2.5. Processing and Data Analysis

The data processing and analysis stages were described in quantitative and qualitative data. The quantitative data were analyzed and processed through cleaning, editing, coding, recoding, and computing using SPSS statistical software version 25. Data on the volume of endoscopy services from January 2020 to September 2021 were displayed descriptively with patient characteristics. The internal and external factors were displayed as mean ± standard deviation or median (range) and percentage on numeric and categorical data. Furthermore, bivariate data in the form of an Independent T-Test (assuming normally distributed data) were analyzed to determine differences in the volume of Endoscopic Services before and during the pandemic. This also includes the relationship between the Pandemic Period, Patient Age and Gender, Type of Patient Case, Origin of Care, Type of Endoscopic Procedure, and Volume of Endoscopic Services. Meanwhile, differences in patient gender characteristics, type of patient case, the origin of care (outpatient or emergency department), and type of endoscopic procedure before and during the pandemic were analyzed using the Chi-Square Test. Multivariate analysis was performed by linear regression to evaluate the number of endoscopic volume changes. The model includes the intercept and slope of each independent factor that contributes as predictors of endoscopic volume. A *p*-value under 0.05 was considered statistically significant. Cohen’s d effect sizes were calculated to compare change procedure endoscopic before and during the COVID-19 pandemic. Qualitative data processing and analysis were undertaken through data reduction, data presentation, conclusion drawing, and verification [11].

## 3. Results

We assessed the different volumes of endoscopic across different time intervals. Period A before the pandemic (January–March 2020) and period B-E (April–September 2021) during the pandemic with a three month interval. The data analyzed showed a significant difference in the mean endoscopy volume, age, number of COVID-19 cases, gender proportion, type of patient case, the origin of care (outpatient or emergency department), and the proportion of types of endoscopic procedures (diagnostic and therapeutic endoscopy, such as band ligation/histoacryl injection, colonoscopy, endoscopic ultrasonography (EUS), endoscopic retrograde cholangiopancreatography (ERCP).

### 3.1. Endoscopic Service Volume Pattern at Cipto Mangunkusumo Hospital (January 2020–September 2021)

The period between January 2020 and September 2021 was divided into January–March 2020 (period A), representing the period before the pandemic occurred. Furthermore, the period after it had occurred includes April–June 2020, as the initial period of the pandemic (period B), July–September 2020 (period C), October–December 2020 (period D), January–March 2021 (period E), April–June 2021 (period F), and July–September 2021 (period G). A total of 5030 volumes of endoscopic procedure services were recorded at Dr. Cipto Mangunkusumo Hospital. The volume pattern of the endoscopy services during these periods is presented in Figure 1 as follows.

Figure 1 above shows that the highest volume of endoscopy services was in period A (January–March 2020), which occurred before the COVID-19 pandemic. In comparison, the period for the lowest volume of endoscopy services was in period B (April–June 2020), at the early stage.

### 3.2. Comparison of Endoscopy Services before and during the COVID-19 Pandemic

#### 3.2.1. Comparison of Average Volume of Endoscopy Services before and during the COVID-19 Pandemic

The average volume of endoscopy services per month at the six intervals of the AG period can be seen in Table 1. It is shown that in period A, before the COVID-19 pandemic, the average number of patients who experienced endoscopic procedures in PESC units was 358 ± 76.1 per month. However, during the pandemic, the number decreased, specifically in period B (the early COVID-19 pandemic), which had 124 ± 35.9 procedures. The decrease in endoscope volume in periods A and B based on the T-test was statistically significant with a *p*-value = 0.009.

An overview of the overall average volume of endoscopy services compared to the period before (period A) and during the pandemic (accumulative average from period B–G) is shown in Figure 2 and Table 2. The average volume of endoscopy services before (January–March 2020) and during the pandemic was 358 ± 76.1 and 219 ± 67.5 per month. Based on the T-test, the decrease in endoscopy volume was statistically significant, with a *p*-value = 0.004.

#### 3.2.2. Estimated the Number of Endoscopic Procedures between 2020–2021 during the COVID-19 Pandemic

The model explains that two independent factors influenced the number of volumes of endoscopic changes during 2020–2021. First, during the pandemic, the endoscopic volume continued to rise but showed a slower decline in the last period, which could be assumed as an effect of the wave secondary pandemic in 2021. The linear regression equation (Table 3) that fit into this model was Y (endoscopic volume)= 384.54 − 5.59 (male) + 9.42 (hematemesis melena). The volume of endoscopic will increase if the proportion of urgency cases (hematemesis melena) and gender (female) are raised.

The value of R Square (Table 4) indicates that this model contains gender (male) and case type (hematemesis melena) that can explain about 57% of endoscopic volume; the rest is affected by other variables.

### 3.3. Comparison of Average Endoscopy Service Volume, Patient Age, Number of COVID-19 Cases, Gender Proportion, Type of Patient Case, Origin of Care, and Type of Endoscopic Procedure before and during the COVID-19 Pandemic

A comparison of endoscopic services volume and the factors involved before and during the COVID-19 pandemic is summarized in Table 5. The study analyzed the difference between Average Endoscopy Service Volume, Patient Age, Number of COVID-19 Cases, Gender Proportion, Type of Patient Case, Origin of Care, and Type of Endoscopic Procedure (histoacryl ligation/injection, colonoscopy, endoscopic ultrasonography/EUS), endoscopic retrograde cholangiopancreatography/ERCP). This was carried out before the pandemic occurred between January–March 2020 (period A) and after it had occurred between April–June 2020 (period B), July–September 2020 (period C), October–December 2020 (period D), January–March 2021 (period E), April–June 2021 (period F), and July–September 2021 (period G). Furthermore, there was no difference in the proportion of patient gender before and during the COVID-19 pandemic. However, there was a difference in the proportion of females who experienced endoscopic procedures, increasing from 39% to 49.9%, which was statistically significant (Chi-Square test *p* < 0.001) only in period F (April–June 2021). However, the proportion of males dominated the volume of endoscopy services in each period.

The Pearson Correlation Test showed no correlation between the volume of endoscopy services and the trend in the number of COVID-19 cases in Jakarta City. The fluctuating volume of endoscopy services during the pandemic is shown in Figure 3. Though the fluctuations in the volume of endoscopy services seem to follow the trend of daily COVID-19 cases in Jakarta, statistically, there was no relationship. Furthermore, there appeared to be a significant difference in the proportion between patient admissions from outpatient and emergency procedures before and during the pandemic. The indication of patients who underwent endoscopy in period A was dominated by non-emergency cases (70.9%). Meanwhile, in the B g period, endoscopic indications caused by non-emergency cases decreased by 57.7–66.3%. The results of the Chi-Square Test also showed that the proportion of patients admitted for outpatient procedures decreased significantly during the B–G period. Overall, 38.2% of patients received endoscopy services, and the proportion of endoscopic services volume originating from outpatients decreased significantly during the pandemic. There was a difference in the types of action taken between periods A and B to G. Generally, the proportion of each action has decreased compared to before the pandemic. The trend that showed significant differences also includes diagnostic esophagogastroduodenoscopy (EGD) and histoacryl injection in varicose veins.

This study combines the criteria for Urgent and Emergency cases. Figure 3 explains the trend in the volume of endoscopy services along with the number of COVID-19 cases in Jakarta per pandemic period.

The change in the flow of action from an outpatient to a complete inpatient unit during the COVID-19 pandemic differed from patients who entered the emergency department. (Figure 4, Figure 5 and Figure 6). The changes were more pronounced in the long waiting room entry queues and the mandatory COVID-19 PCR swab. This is because patients admitted through the emergency department are usually classified as urgent/emergency cases, such as hematemesis melena (upper/lower gastrointestinal bleeding), jaundice (icteric), and other concomitant conditions. As a result, the patient could not be treated in a special ward for a fast-track procedure.

### 3.4. Changes in the Flow of Endoscopic Procedures during the COVID-19 Pandemic

Based on the Standard Operational Procedure in Dr. Cipto Mangunkusumo Hospital regarding the administration of inpatients with elective surgery, quick-track, chemotherapy, and patients with internal medicine procedures: “Fast-Track procedure is an immediate patient action process that does not require long-term treatment but involves a series of examinations which are carried out at the outpatient polyclinic. This is needed to provide certainty regarding the scheduling of actions, ensure the smooth flow of inpatients and fulfill safety standards.” (Dr. Cipto Mangunkusumo Hospital, 2018). The case manager’s job is to maintain quick-track service administration, verify the inpatient application file (SPR), complete the coding for outpatient action tools, and create treatment plans. However, there were only five rooms (30 beds) available for fast-track procedures in the hospital.

Preparations from before, during, and after endoscopic procedures in preventing COVID-19 infection follow the guidelines from national and international gastroenterology–hepatobiliary associations and the Indonesian Ministry of Health. However, the service flow became more prolonged during this process, and patients experienced queues before entering the operating room. This was because the bed capacity was not sufficient for all procedures in the internal medicine ward, which does not only accept endoscopic patients. In addition, sometimes, patients’ laboratory examinations must be repeated because they had expired when they arrived in the room.

The decrease in endoscopy volume was due to management policies and the agreement of the doctor itself to limit the number of patients seeking treatment in the outpatient unit and the number of procedures for those with non-urgent/emergency cases. However, the doctors never lowered the quality of endoscopy services. On the contrary, the duration of the action procedure was accelerated without reducing its quality. Moreover, various efforts are still being made to protect patients and all personnel in the procedure room. The assessment of service quality through the level of patient satisfaction during the pandemic has not been carried out formally. However, based on the doctors’ opinions, they are pretty understanding about various changes in the flow of services and preparations for health actions to prevent COVID-19 infection.

## 4. Discussion

### 4.1. Endoscopy Volume Changes during the COVID-19 Pandemic

This study showed a decrease in the cumulative volume of endoscopic procedures, diagnostic and therapeutic upper GI endoscopy, diagnostic and therapeutic lower GI endoscopy, ERCP, dan EUS during the pandemic, specifically in period A (January–March 2021). The decrease in volume was very high in period B (April–June 2020) by 65%, specifically at the early onset of the pandemic. It also occurred in period C (July–September 2020) by 18%, period D (October–December 2020) by 40%, period E (January–March 2021) by 28%, period F (April–June 2021) by 13%, and period G (July–September 2021) by 40%. This was because there was no preparedness regarding prevention and control policies, specifically in endoscopy facilities. In addition, the second-largest decline occurred in period G (the transmission period of the Delta variant). The most significant endoscopy volume increase occurs from period B to period C. In the national situation, the ministry declaration of largescale social restrictions (LSSR) or *Pembatasan Sosial Berskala Besar* (PSBB) was the first response to block the transmission of COVID-19. The impact of the first PSBB was to restrict the activity and mobility of society by up to 100%. The implementation of PSBB in Indonesia seems to strongly reduce the average number of endoscopic volume in period B. PSBB transition in line with period C (July–September 2020), with the activity and mobility of society restriction stretched to 75%. Further, relaxation in the PSBB transition increases the endoscopy volume by bringing the patients to return to the hospital.

In general, the decrease in patient visits to the hospital and the volume of endoscopic procedures was due to the policy of limiting visits at the early onset of the pandemic from the hospital management and the doctors. The decline in endoscopic volume in period G (July–September 2021) was not too drastic and almost resembled the average volume in other periods during the pandemic. An increase follows this in the daily number of COVID-19 cases in Jakarta. Furthermore, this change is consistent with some literature showing a decrease in endoscopic volume by 60% from about 80% of endoscopy centers worldwide. After comparing the volume of endoscopic procedures before and during the COVID-19 pandemic, it was found that there was an 80% reduction in esophagogastroduodenoscopy (*p* < 0.0001), colonoscopy (*p* < 0.0001), capsule endoscopy (*p* < 0.0001), and manometric procedures (*p* = 0.002). The mean number of ERCP procedures was statistically significant (*p* < 0.0001) and experienced the lowest percentage reduction among all endoscopes [12]. Latina et al. (2021) showed that the volume of gastroscopy (57%) and colonoscopy procedures (45%) decreased during the lockdown period in the Netherlands. However, the volume of endoscopic retrograde cholangiopancreatography remained the same and also returned to normal after the lockdown [13]. 

The results also showed that the number of volumes of endoscopic changes during 2020–2021 was influenced by the effect of the secondary wave pandemic in 2021, gender (female), and the proportion of urgency cases (hematemesis melena) as much as 57%. This is in line with the study by Ying et al. (2022), which showed that patients who underwent endoscopy during the pandemic had higher proportions of urgent cases caused by hospital patients’ admission limitation policy or reluctance to seek hospital care during a pandemic [14]. However, the fact that this study results had female higher proportions than the male who underwent endoscopic procedures during the pandemic is contrary to a previous study from Ugurlu (2021), which mentioned that the majority of the patient who underwent the endoscopic procedure were men before and after the pandemic [15]. A large-population-based study from Srinivasan (2021) also showed that women respondents were significantly more likely NOT to undergo an endoscopy procedure during the pandemic (1.16; 95%CI 1.09–1.23) [16]. This may depend on each hospital’s situation. The effect of the secondary wave pandemic in 2021 made the decreased endoscopic procedures much slower than the first wave of the pandemic, the same as a study from Kuo Tony et al. (2022), who mentioned that after the second wave of the pandemic, the public fear of COVID-19 reduced slightly and began to seek health care again [17].

### 4.2. Comparison of Average Endoscopy Service Volume, Patient Age, Number of COVID-19 Cases, Proportion of Gender, Type of Patient Case, Origin of Care, and Type of Endoscopic Procedure before and during the COVID-19 Pandemic

The results of our study show that various factors such as age, gender, type of patient case (a diagnosis that is an indication for endoscopy), the origin of treatment, and type of endoscopic procedure performed significantly differed during the COVID-19 pandemic compared to before the pandemic. There was a decrease in the average age of patients undergoing endoscopic procedures during the COVID-19 pandemic, which was 47 years compared to 50 years before the pandemic. This may also be related to the government’s appeal to people with comorbidities, the elderly, or even pre-elderly age (50–59 years) to avoid traveling out of the house or visiting the hospital when necessary [18].

The gender factor was not statistically significant. However, in general, the proportion between male and female patients in each period is always higher. This is also related to the predilection for chronic liver disease or liver cirrhosis, who routinely perform periodic endoscopies to screen the esophagus. It is found in men, so although the number has decreased, the proportion of male patients is more than female patients [19].

The number of cases of patients undergoing endoscopic procedures has also decreased statistically. However, it can be seen that emergency and urgent cases will not decrease drastically. A drastic decrease, especially in only diagnostic cases or screening and not an emergency or urgent. This is in line with the recommendations and guidelines of the world and local endoscopic societies in countries that recommend efforts to limit patients, especially at the beginning of the pandemic and when cases are soaring high. The recommendation only recommends performing endoscopic measures in emergency or urgent cases. However, when cases began to decline, and policies and measures for screening and infection prevention were improved, the restrictions on cases undergoing endoscopic procedures were also relaxed [20,21].

The origin of patient care indicates the origin of the patient’s entrance to the procedure. Patients can be admitted from the outpatient department or the emergency department. Before the COVID-19 pandemic, patients could undergo endoscopic procedures from an outpatient unit. However, after the pandemic, all patients undergoing endoscopic procedures must be hospitalized to facilitate various screening preparations before the procedure. This made a statistically significant difference in the mean number of patients admitted from the outpatient unit and the emergency department. The mean of outpatients decreased drastically, while those admitted from the emergency department were not significantly different. The cases from the emergency department are emergency or urgent so that they will not be too different from before the pandemic. Although it can be seen that there has also been a drastic decrease of around 50% from the previous cases entering the emergency unit, especially at the beginning of the pandemic and during the G period when there was a spike in cases due to the Delta variant [21].

This endoscopic procedure is also related to the selection of emergency and urgent cases, which are more therapeutic than diagnostic. This can be seen from the drastic decline in diagnostic endoscopic procedures, such as EGD, colonoscopy, or EGD and colonoscopy. Meanwhile, other therapeutic measures, such as esophageal variceal ligation, histoacryl injection, ERCP, and EUS, even though the reduction will not be too drastic. This is in line with research results in other countries, which show that therapeutic actions in emergency and urgent cases will still be carried out [21,22].

The results of this study clearly showed that there are statistically significant differences between the mean endoscopy volume, patient age, type of patient case, origin of care as the patient’s entry point, and type of endoscopic procedure between before (period A) and during the COVID-19 pandemic (period B-G). However, the proportion of gender and the number of daily COVID-19 cases in Jakarta are not statistically too different. In general, the gender of the patient is associated with a predilection for gastro–hepatic diseases, especially liver cirrhosis, which is more common in men. So even during the COVID-19 pandemic, the proportion of males will still be higher.

### 4.3. Changes in the Flow of Endoscopic Procedures during the COVID-19 Pandemic

Changes in the flow of services that occur for the screening process and infection prevention have caused various other impacts on services, such as: (1) The journey that the patient has to go through becomes longer and longer; (2) patient queues for procedures will be longer and patient procedures will be delayed for a long time because patients must be treated at the time of the procedure, while the number of rooms for internal medicine procedures (shared with other internal medicine procedures) is only five rooms or 30 bed; (3) the repetition of laboratory examinations and PCR swabs will be repeated because when a new patient receives a treatment room, the results needed for the procedure have expired and need to be repeated; (4) treatment in the fast track room that ideally only takes 1–2 days will be prolonged because the patient still has to repeat laboratory results or PCR swabs, which also cannot be performed every day (there are specific schedules, especially during holidays or weekends they cannot be performed). Several problems arising from this change in the flow of services must be evaluated further because it will impact patient care and financing.

### 4.4. Lesson-Learned from the Study

This study is indeed a cross-sectional case study in one hospital. However, the hospital used as the study location is Indonesia’s number one national referral hospital. It can be a model for other hospitals throughout Indonesia regarding hospital governance, clinical management, cost accounting, policy systems, implementation, and preparation. Additionally, its response in the face of the COVID-19 pandemic.

Some recommendations that might be an initial solution to overcome various problems that arise in the field related to hospital readiness in dealing with the future pandemic risk are as follows: (1) It is necessary to make SOP or uniformity of service flow in work units that provide the same service product, even though this COVID-19 infection is an example of a very dynamic and unstable situation. However, the absence of standardized standards can be a gap in providing services that are not up to standard. There is at least a clinical practice guideline for procedures for infectious infections. Next, starting to separate infectious and non-infectious patients by utilizing Kiara’s treatment room (a former COVID-19 treatment room), although still providing bed reserves for the possibility that COVID-19 cases will rise again. This separation effort will undoubtedly increase the ward’s capacity for patients undergoing procedures, including endoscopy.

Meanwhile, all patients undergoing endoscopic procedures remain in the hospital first while conducting a study to analyze the cost benefits of performing outpatient and inpatient endoscopic procedures. Furthermore, it is necessary to think about efficiency efforts in conducting laboratory examinations and PCR swabs so that they are not carried out repeatedly, for example, by doing only when the patient has entered the room. The blood examination is made, so the results can come out in a few hours and immediately schedule an action for the patient or perform specific treatments if the laboratory results show blood products that need to be added before the procedure.

Additionally, it is necessary to immediately conduct a study to calculate the unit cost of endoscopic procedures by changing the service flow and adding various preparations to deal with COVID-19. In addition, efforts to implement, monitor, and evaluate policies must be socialized to the lowest level and require the most effective method of socialization, for example broadcasting private messages to each employee from the system and periodically providing a reminder system and monitoring its implementation by the head of each unit.

This study has a limitation that needs to be concerned. Although the study had a significant effect size, which means that the findings of endoscopic procedures have practical significance deceased before and during the COVID-19 pandemic, the sample size was only from one tertiary hospital centre, and the results might not be generalizable in other centers.

## 5. Conclusions

The pattern of decreasing endoscopy volume before and during the COVID-19 pandemic showed a statistically significant difference. It was very drastic in period B (the beginning of the pandemic) and period G (when there was a spike in cases due to the Delta variant) related to case restriction and management measures. Moreover, the doctor in charge and hospital readiness factors in implementing various efforts to prevent COVID-19 infection.

This difference in endoscopy volume was also accompanied by statistically significant differences in the patient’s age, the type of case that indicated the patient underwent endoscopy, the origin of treatment, and the type of endoscopic procedure performed according to the initial hypothesis proposed and from previous studies.

Various impacts of changes in service flow need to be followed up because they can cause new, more extensive problems, so solutions must be immediately sought, or study efforts must be made to find the right solution. First, it is necessary to make SOP or uniformity of service flow in work units that provide the same service product, even though this COVID-19 infection is an example of a very dynamic and unstable situation. There should be at least a clinical practice guideline for procedures for infectious infections. Next, starting to separate infectious and non-infectious patients. This separation effort will undoubtedly increase the ward’s capacity for patients undergoing procedures, including endoscopy. Efforts to socialize policies from the highest level to the bottom require simultaneous and continuous efforts, and there must be an evaluation of whether each party has accepted the policy and how it is implemented. This research is cross-sectional with case studies at the number one national referral hospital in Indonesia, but of course, it will not necessarily show the same results in various other referral hospitals in Indonesia.

Further research is needed to assess other outcomes of services, such as calculating unit costs of measures during a pandemic, patient satisfaction levels, and cost benefits of outpatient and inpatient measures. However, the results of this study can be used as a basis for making improvement efforts starting from the lowest work unit, namely the endoscopy unit, as an initial model for making SOPs in general for endoscopy in cases of infection.

It is necessary to conduct a similar study in other endoscopic referral hospitals with a better prospective cohort method to minimize the weaknesses in this study, but if not possible, at least use the same method. The research basis from these hospitals can later be combined to be proposed to improve existing policies related to health services and financing.

## Figures and Tables

**Figure 1 healthcare-10-02280-f001:**
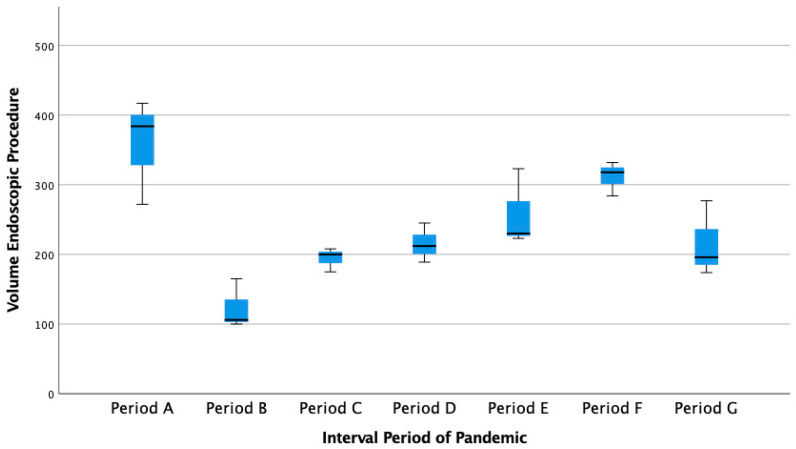
The pattern of Endoscopic Service Volume based on the COVID-19 Pandemic Interval Period.

**Figure 2 healthcare-10-02280-f002:**
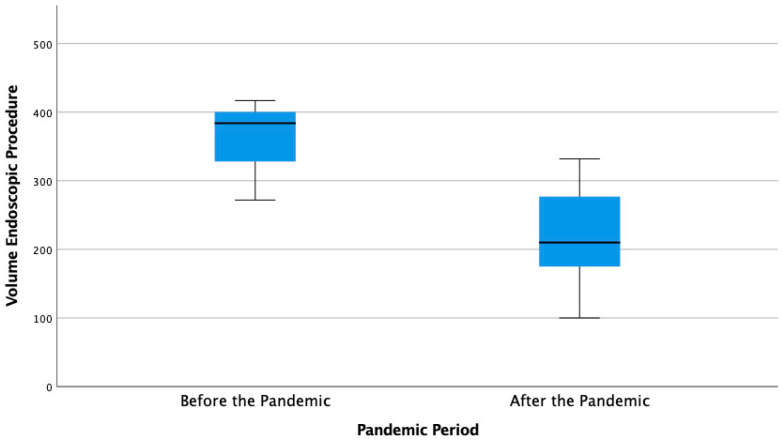
Comparison of Average Accumulative Volume of Endoscopic Services before and During the Pandemic.

**Figure 3 healthcare-10-02280-f003:**
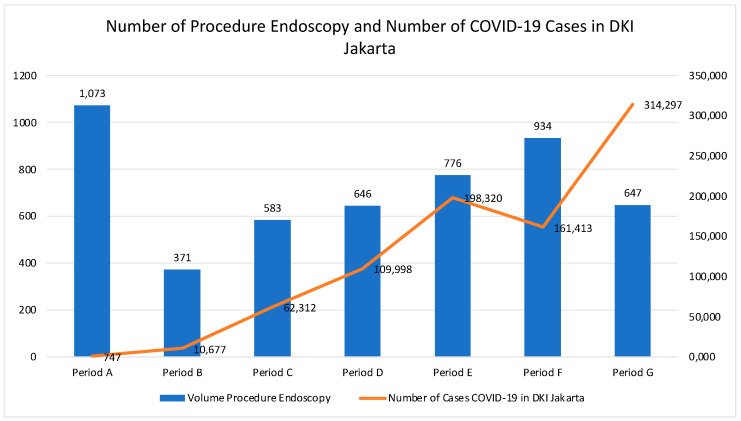
Trends in Endoscopy Service Volume Compared to Daily COVID-19 Cases in Jakarta.

**Figure 4 healthcare-10-02280-f004:**
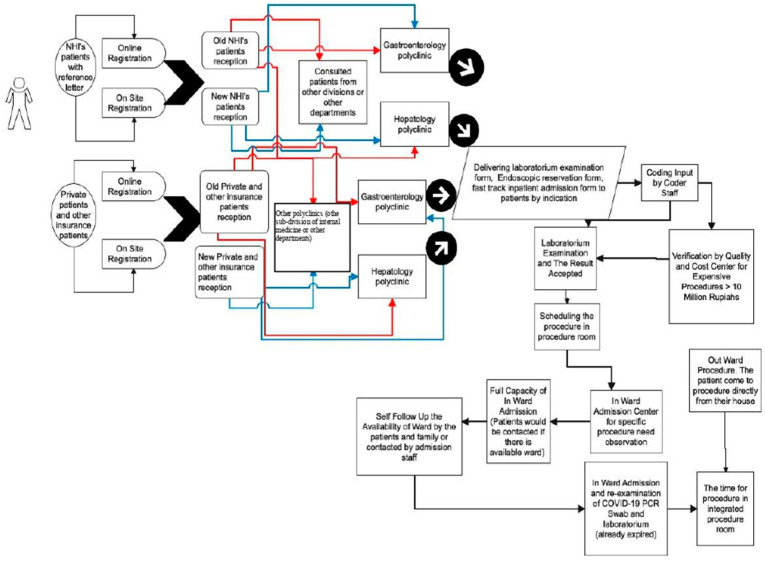
Entry Flow of Endoscopic Patients from Outpatient Units before the Pandemic.

**Figure 5 healthcare-10-02280-f005:**
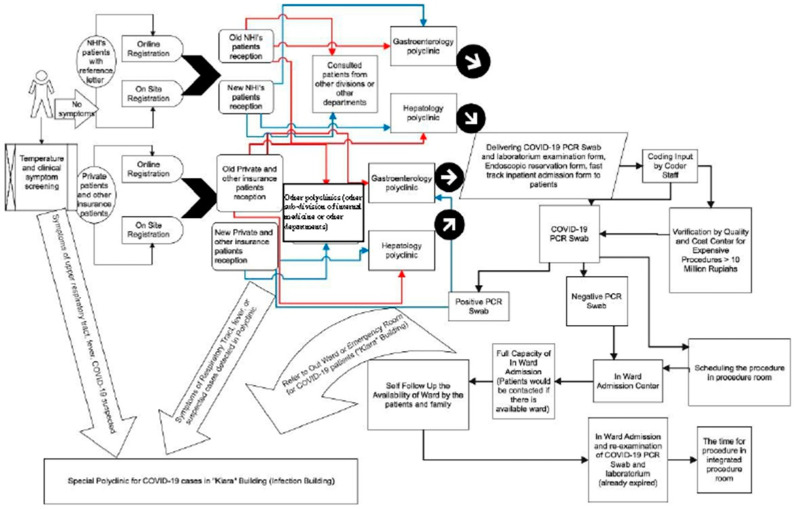
Entry Flow of Endoscopic Patients from Inpatient Units before the Pandemic.

**Figure 6 healthcare-10-02280-f006:**
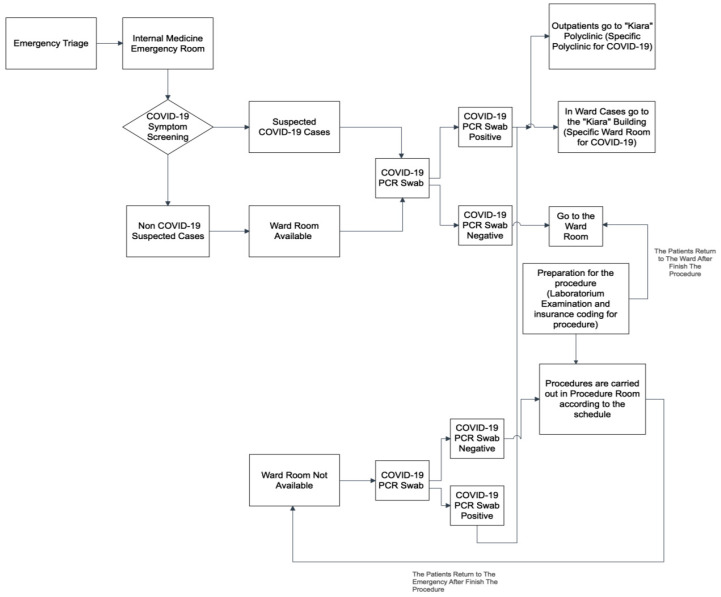
Entry Flow of Endoscopic Patients from the Emergency Room During the Pandemic.

**Table 1 healthcare-10-02280-t001:** Average Volume of Endoscopic Services in Periods A to G.

Endoscope Volume per Month	Period A	Period B	Period C	Period D	Period E	Period F	Period G
Mean ± Standard Deviation	358 ± 76.1	124 ± 35.9	294 ± 17.2	215 ± 28.1	259 ± 55.8	311 ± 24.7	216 ± 54.2
*p*-value	reff	0.009	0.022	0.038	0.143	0.372	0.058
Cohen’s d	reff	3.936	2.964	2.484	1.479	0.825	2.146

**Table 2 healthcare-10-02280-t002:** Average Accumulative Volume of Endoscopic Services before and During the COVID-19 Pandemic.

Endoscope Volume per Month	Before the COVID-19 Pandemic	During the COVID-19 Pandemic	*p*-Value	Cohen’s d
Mean ± Standard Deviation	358 ± 76.1	219 ± 67.5	0.004	1.917

**Table 3 healthcare-10-02280-t003:** Regression Coefficient.

Parameters	Estimate Coefficient Beta	Standard Error	T Value	*p* Value
Intercept	384.54	172.25	2.23	0.041
Gender (male)	−5.59	2.65	-2.11	0.052
Case type (Hematemesis Melena)	9.42	3.00	3.14	0.007

**Table 4 healthcare-10-02280-t004:** R Square value.

R	R Square	Adjusted R Square	Standard Error
0.755	0.570	0.513	47.132

**Table 5 healthcare-10-02280-t005:** Comparison of Endoscopy Volume, Patient Age, Number of COVID-19 Cases, Gender Proportion, Type of Patient Case, Origin of Care, and Type of Endoscopic Procedure before and During the COVID-19 Pandemic.

	Period A	Period B	Period C	Period D	Period E	Period F	Period G
Endoscope Volume per Month, Mean ± Standard Deviation	358 ± 76.1	124 ± 35.9	294 ± 17.2	215 ± 28.1	259 ± 55.8	311 ± 24.7	216 ± 54.2
*p*-value (Independent T-Test)	Reff	0.009	0.022	0.038	0.143	0.372	0.058
Patient Age, Mean ± Standard Deviation	50 ± 17.4	47 ± 17.3	48 ± 17.3	47 ± 17.6	47 ± 17.8	47 ± 17.5	47 ± 18.6
*p*-value (Independent T-Test)	Reff	0.009	0.023	< 0.001	0.001	0.001	0.001
Number of COVID-19 Cases, Mean ± Standard Deviation	747	3559 ± 465.5	20,770 ± 11,974.3	36,666 ± 8845.4	66,106.7 ± 22,092.6	53,804 ± 51,682.7	104,765 ± 144,838.4
Correlation r value	-	−0.808	−0.868	−0.916	−0.561	−0.200	−0.785
*p*-value (Correlation Test)	Reff	0.192	0.132	0.084	0.439	0.8	0.215
Patient Gender, *n* (%)							
-Male	654 (61%)	226 (60.9%)	332 (56.9%)	395 (61.1%)	454 (58.5%)	468 (50.1%)	358 (55.3%)
-Female	419 (39%)	145 (39.1%)	251 (43.1%)	251 (38.9%)	322 (41.5%)	466 (49.9%)	289 (44.7%)
*p*-value (Chi-Square Test)	Reff	1	0.125	0.977	0.312	<0.001	0.03
Patient Case Type, *n* (%)							
-Non-Emergency							
-Urgent/Emergency							
a. Upper Gastrointestinal Tract							
**Hematemesis Melena**	761 (70.9%)	246 (66.3%)	354 (60.7%)	385 (59.6%)	448 (57.7%)	567 (60.7%)	399 (61.7%)
b. Biliary System							
**Icterus Obstruktif**	133 (12.4%)	42 (11.3%)	91 (15.6%)	106 (16.4%)	134 (17.3%)	181 (19.4%)	115 (17.8%)
c. Lower Gastrointestinal Tract	122 (11.4%)	54 (14.6%)	90 (15.4%)	81 (12.5%)	96 (12.4%)	90 (9.6%)	85 (13.1%)
**Hematochezia**	57 (5.3%)	29 (7.8%)	48 (8.2%)	74 (11.5%)	98 (12.6%)	96 (10.3%)	48 (7.4%)
*p*-value (Chi-Square Test)	Reff	0.021	<0.001	<0.001	<0.001	<0.001	0.001
Origin of Treatment							
-Emergency	663 (61.8%)	321 (86.5%)	553 (94.9%)	624 (96.6%)	667 (86.0%)	680 (72.8%)	531 (82.1%)
-Outpatient	410 (38.2%)	50 (13.5%)	30 (5.1%)	22 (3.4%)	109 (14%)	254 (27.2%)	116 (17.9%)
*p*-value (Chi-Square Test)	Reff	<0.001	<0.001	<0.001	<0.001	<0.001	<0.001
Type of Procedure, *n* (%)							
-EGD	533 (49.7%)	161 (43.4%)	199 (34.1%)	248 (38.4%)	292 (37.6%)	388 (41.5%)	251 (38.8%)
-VE ligation	45 (4.2%)	9 (2.4%)	23 (3.9%)	37 (5.7%)	17 (2.2%)	37 (4.0%)	15 (2.3%)
-Colonoscopy	203 (18.9%)	62 (16.7%)	113 (19.4%)	107 (16.6%)	146 (18.8%)	187 (20.0%)	136 (21.0%)
-Histoacryl Injection	14 (1.3%)	3 (0.8%)	13 (2.2%	6 (0.9%)	3 (0.4%)	1 (0.1%)	0 (0%)
-EUS	28 (2.6%)	16 (4.3%)	34 (5.8%)	30 (40.6%)	54 (7.0%)	52 (5.6%)	38 (5.9%)
-ERCP	130 (12.1%)	64 (17.3%)	98 (16.8%)	85 (13.2%)	104 (13.4%)	108 (11.6%)	94 (14.5%)
-EGD + VE ligation	64 (6.0%)	29 (7.8%)	56 (9.6%)	60 (9.3%)	64 (8.2%)	67 (7.2%)	52 (8%)
-EGD + Colonoscopy	56 (5.2%)	27 (7.3%)	47 (8.1%)	73 (11.3%)	96 (12.4%)	94 (10.1%)	61 (9.4%)
*p*-value (Chi-Square Test)	Reff	0.017	<0.001	<0.001	<0.001	<0.001	<0.001

## Data Availability

Not appliable.

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
