# Peer review of "Factors Influenced the Endoscopic Services Volume during the COVID-19 Pandemic at National Tertiary Referral Hospital in Indonesia: Dr. Cipto Mangunkusumo Hospital"

_healthcare, 2022, doi:10.3390/healthcare10112280_

Round 1
Reviewer 1 Report (Previous Reviewer 2)
Thank you for the response and revisions.
Author Response
Thank you for the comments and support

Reviewer 2 Report (New Reviewer)
This is a large volume study of endoscopic service during the Covid 19 pandemic.
The readability and grammar of English writing may benefit from editing by a native.
The volume change is well-described.
The most interesting finding perhaps is shown in Table 5. There appears to be no demonstrative change of the types of pathologies (i.e. increase or decrease in percentages) in upper and lower GI bleeding or obstructive jaundice. This point may be further highlighted in discussion, as it suggests Covid infection has much less or no damage on GI tract than respiratory system. However, it may be worthwhile to find out the increased emergency endoscopy procedures were due to the arrangement during the pandemic or due to the urgency of bleeding.
Younger population in the cohort is another interesting point worth discussing. Is it due to altered immune reaction in the younger population leading to bleeding or social behavior?
If data are available, it would also be interesting to correlate Covid status (positive or negative) with each type of pathologies
In the discussion, one can also elaborate on how the findings of this study prepare the hospital administration prepare for future pandemics like Covid.
The conclusion should highlight
Author Response
This is a large-volume study of endoscopic service during the COVID-19 pandemic. The readability and grammar of English writing may benefit from editing by a native.
Thank you very much for the suggestion. We have used a native translator for preparing this manuscript.
The volume change is well-described.
Thank you very much for the comment.
The most exciting finding, perhaps, is shown in Table 5. There appears to be no demonstrative change in the types of pathologies (i.e., increase or decrease in percentages) in upper and lower GI bleeding or obstructive jaundice. This point may be further highlighted in the discussion, as it suggests COVID-19 infection has much less or no damage to the GI tract than the respiratory system. But, it may be worthwhile to determine whether the increased emergency endoscopy procedures were due to the arrangement during the pandemic or the urgency of bleeding.
Thank you very much for the suggestions. We have highlighted the point about the percentages of each intervention in gastrointestinal cases which were not influenced by COVID-19 infection. All the interventions collected in this study were negative COVID-19 swab PCR. We did not include the procedures performed on COVID-19 patients because those cases were exceptional. The determination of issues for endoscopic procedures is based on the case's urgency because there are several endoscopic procedures during the COVID-19 pandemic, so cases handled for endoscopic procedures during this period are only indeed for therapeutic cases (not for diagnostic procedures).
The younger population in the cohort is another interesting point worth discussing. Is it due to altered immune reactions in the younger population leading to bleeding or social behavior?
The younger population who underwent an endoscopy procedure in this study was caused by social behavior where the younger population was more willing to go to the hospital because of complaints of illness than the elderly population during the pandemic period. This may also be related to the government's appeal to people with comorbidities, the elderly, or even pre-elderly age (50-59 years) to avoid traveling out of the house or visiting the hospital when necessary.
If data are available, it would also be interesting to correlate Covid status (positive or negative) with each type of pathologies
We excluded the positive COVID-19 patients for this research. This research wants to show the regular services in endoscopy.
In the discussion, one can also elaborate on how the findings of this study prepare the hospital administration for future pandemics like COVID-19.
Some recommendations that might be an initial solution to overcome various problems that arise in the field related to hospital readiness in dealing with the future pandemic risk are as follows: 1) It is necessary to make SOP or uniformity of service flow in work units that provide the same service product, even though this COVID-19 infection is an example of a very dynamic and unstable situation. However, the absence of standardized standards can be a gap in providing services that are not up to standard. There is at least a clinical practice guideline for procedures for infectious infections. Next, starting to separate infectious and non-infectious patients by utilizing Kiara's treatment room (a former COVID-19 treatment room), although still providing bed reserves for the possibility that COVID-19 cases will rise again. This separation effort will undoubtedly increase the ward's capacity for patients undergoing procedures, including endoscopy.
The conclusion should highlight
The conclusion has added some recommendations regarding hospital administration management for pandemic situations.

This manuscript is a resubmission of an earlier submission. The following is a list of the peer review reports and author responses from that submission.
Round 1
Reviewer 1 Report
This manuscript aims to examine changes in endoscopic services to identify service volume patterns and influence factors quantitatively and qualitatively at one hospital in Indonesia during the 82 COVID-19 pandemic between 2020 to 2021.
The main problem with this work is that it uses a pre-COVID-19 period for comparing the volume of endoscopic services January-March 2020, just the 3 months immediately before the pandemic began. To what extend the volume of cases during this very short period is really representative of the case-load of that hospital is not clear.
Another problem with this manuscript is the methods proposed to compare the before-after volume of cases: correlations and chi-square and the possible effect of contributing factors are analyzed independently making it difficult to have a define picture of the specific independent effect of each of them.
Author Response
Dear Reviewer,
Thank you very much for the review and suggestions to improve this manuscript. We have tried to revise some issues as suggested.
This manuscript aims to examine changes in endoscopic services to identify service volume patterns and influence factors quantitatively and qualitatively at one hospital in Indonesia during the 82 COVID-19 pandemics between 2020 to 2021.
Response :
We have mentioned it in the introduction.
The main problem with this work is that it uses a pre-COVID-19 period for comparing the volume of endoscopic services January-March 2020, just the three months immediately before the pandemic began. So, to what extent the volume of cases during this brief period is representative of that hospital's caseload is unclear.
Response :
Revised 2.3. Population and Samples.
The volume of endoscopic was divided into periods A-G with three months intervals from January 2020 until September 2021. The sample minimum for the volume of endoscopic services before the pandemic (January – March 2020) and during the pandemic (April 2020 – September 2021) was calculated by unpaired numerical comparative test formula resulting in a total number of minimal 48 patients in each period A-G. Chai et al. report that there were significant differences in GI endoscopic procedure between 2019 and 2020 in the period January – April, hence we selected three months as a time interval by considering the previous studies that were used as a reference for sample calculations, the load cases COVID-19 in Jakarta that were spreading fast, and the response of hospital in preparing for adaptation regulations.
In period A, before the COVID-19 pandemic, the average number of patients who underwent endoscopic procedures in PESC and RPT units was 358 procedures per month, totaling 1,073 cases. This number is sufficient to represent the total caseload in the hospital because, in general, PESC and RPT units can perform endoscopic procedures up to 1,000 – 2,000 cases per month. While during the COVID-19 pandemic, the number of these patients decreased, with a significant decrease in period B (beginning of the COVID-19 pandemic) which was 124 cases (total of 371 points). The decline in endoscope volume in periods A and B was also statistically significant, with a p-value of 0.009.
Reference: Chai N, Tang X, Linghu E, Feng J, Ye L, Wu Q, Zhao X, Du R, Li L, Zhang W, Xiang J; COVID-19 Working Group of the Chinese Society of Digestive Endoscopy. The influence of the COVID-19 epidemic on the gastrointestinal endoscopy practice in China: a national survey. Surg Endosc. 2021 Dec;35(12):6524-6531. DOI: 10.1007/s00464-020-08149-4. Epub 2020 Nov 11. PMID: 33179181; PMCID: PMC7657378.
Another problem with this manuscript is the methods proposed to compare the before-after volume of cases: correlations and chi-square and the possible effect of contributing factors are analyzed independently, making it difficult to have a defined picture of the specific independent impact of each of them.
Response :
This study explained the factors associated with endoscopic volume in a descriptive. However, since the number of time observations is short (5 periods from A), linear regression for predicting the volume endoscopic can't be performed. Hence, this study only explained the data in descriptive for each period interval A - E.
The analysis was carried out using an Independent T-Test because there was no different intervention treatment before and after the pandemic. When using the Dependent T-Test, there needs to be an intervention and a distinction between the control variable group and the dependent group. Although this study compared the conditions before and after the COVID-19 pandemic, internal and external factors such as Patient Age, Patient Gender, Patient Case Type, Origin of Care, and Type of Endoscopic Procedure with Volume of Endoscopic Services could not be analyzed using the dependent test.
Reviewer 2 Report
The authors try to describe the patterns of endoscopic services volume in COVID and the factors associated with this change. The observation and findings are not surprising and have limited implication. If the authors could add information on the effect of decreased visit on treatment delay or disease progression and further stratified by type of disease and gender, etc., the study could have important implications. The following points can be improved for further considerations.
1) The statement in intro should be specified, this reflects the situation in Indonesia or globally? This implies the significances and potential gaps that this study tries to fill in. "The impact of the COVID-19 pandemic caused a decrease in the number of 57 outpatient visits by 60% around early April 2020, and there was also a sharp decline in 58 hospital revenues"
2) The second paragraph in the 2.1. Study Design should be moved to results. And the first sentence seems to be too long, and very hard to read. It's better if you can mention what you did first, for example, "We assessed the difference of A, B, C, D, E across different time intervals. "
3) It is not clear on 2.4. Data Collection Techniques. What is the difference between secondary and primary basis. And what is primary basis?
4) Is there any reason for splitting the whole study period by 3 months?
5) What is the interpretation of the increase from period B to period C? Especially when we see a significant increase of COVID case from period B to C, all the way to G. Even with that high number of cases, the most significant increase occurs at period B.
6) Overall the findings are not too surprising, and what readers probably want to see more is whether this decrease is associated with any delay of treatment, or any progression of disease. Answering this question could have important implications. Can you add more information and analyses results for this part?
Author Response
Dear Reviewer,
Thank you very much for the review and suggestions to improve this manuscript. We have tried to revise some issues as suggested.
The authors try to describe the patterns of endoscopic services volume in COVID and the factors associated with this change. The observation and findings are not surprising and have limited implications. If the authors could add information on the effect of decreased visits on treatment delay or disease progression and further stratified by type of disease and gender, etc., the study could have significant implications. The following points can be improved for further consideration.
The statement in the intro should be specified; this reflects the situation in Indonesia or globally? This implies the significance and potential gaps that this study tries to fill in. "The impact of the COVID-19 pandemic caused a decrease in 57 outpatient visits by 60% around early April 2020, and there was also a sharp decline in 58 hospital revenues."
Response :
This sentence represents globally based on a series of studies from the Commonwealth Fund Foundation in 2020, which stated that the number of visits to ambulatory care providers had declined by nearly 60 percent. However, visits had rebounded by May, though they were still below the pre-pandemic baseline.
References: Ateev Mehrotra et al., The Impact of COVID-19 on Outpatient Visits in 2020: Visits Remained Stable, Despite a Late Surge in Cases (Commonwealth Fund, Feb. 2021). https://doi.org/10.26099/bvhf-e411
The second paragraph in 2.1. The study Design should be moved to results. And the first sentence seems too long and very hard to read. It's better to mention what you did first, for example, "We assessed the difference of A, B, C, D, E across different time intervals. "
Response :
The second paragraph in section 2.1 Study Design has moved to the results on lines 192-193 with a simplified modification of the time interval of the pandemic period. Explanation of the distribution of the pandemic period in section 2.2 Study Location and Time.
It is not clear on 2.4. Data Collection Techniques. What is the difference between a secondary and primary basis? And what is the primary basis?
Response :
Data collection techniques were carried out on a secondary and primary basis. Secondary data on the volume of endoscopy services and patient characteristics, types of patient cases, the onset of treatment, and endoscopic procedures were collected. It was carried out by tracing and retrieving patient data in the hospital's Electronic Health Record (EHR) System. Furthermore, quantitative data collection relating to the number of COVID-19 cases in Jakarta was obtained through the open access report on the "Kawal" website.
Add to paragraph 2.4 in the last paragraph. Data Collection Techniques.
Primary data for qualitative analysis about the comparison of service flow and implementation of standard operating procedures before and during the COVID-19 pandemic was carried out by observation, review documents, and interviews with nurses in the procedure room, head of non-surgical intervention endoscopic service installations, director of administration and finance.
The difference between the secondary database and the primary database is: in the secondary database, the data used is already available at the hospital where the study is located and is not obtained from the results of instruments made by the researcher, such as data on COVID-19 cases, data on mobility restrictions and policies. Patient quota in the hospital service environment; comparison of service flow and implementation of standard operating procedures (SOPs) in endoscopic procedures between before and during the COVID-19 pandemic; and its relation to endoscopy volume during the pandemic was obtained through observation. While the primary database is a primary data collection method in which researchers collect their data through a document review process and in-depth online interviews with related parties using a questionnaire instrument developed by the research team.
Is there any reason for splitting the whole study period by three months?
Response :
Already been revised in 2.3. Population and Samples. This study divides each period with an interval of 3 months because three months is considered an effective period divider to see the significance of both the decrease in endoscopy volume numbers and the increase in COVID-19 cases. In addition, in measuring performance indicators in the endoscopy unit, dr. Cipto Mangunkusumo Hospital is also conducted quarterly.
What is the interpretation of the increase from period B to period C? Especially when we see a significant increase of COVID cases from period B to C, all the way to G. However, even with that high number of cases, the most significant increase occurs in period B.
Response :
Already added in discussion 4.1 Endoscopy Volume Changes During the COVID-19 Pandemic.
The most significant endoscopy volume increase occurs from period B to period C. In the national situation, the government had declared the largescale social restrictions (LSSR) or Pembatasan Sosial Berskala Besar (PSBB) as the first response to block the transmission of COVID-19. The impact of the first PSBB was to restrict society's activity and mobility by up to 100%. As a result, the implementation of PSBB in Indonesia seems to strongly reduce the average number of volume endoscopic in period B. PSBB transition in line with period C (July – September 2020), with the activity and mobility of society restriction stretched to 75%. Further, relaxation in the PSBB transition increases the endoscopy volume by bringing the patients to return to the hospital.
The increase in COVID-19 cases from period B to C is the first increase in cases at the beginning of the COVID-19 pandemic. Although it was not the highest increase during the six pandemic periods (because the highest growth occurred in periods F to G), during this period, the total volume of endoscopes began to increase again after previously dropping drastically from periods A/pre-pandemic to period B. This indicates that, despite a drastic decline, the hospital is slowly trying to increase the number of endoscopy volumes by preparing infection prevention SOPs, reassessing endoscopic emergency criteria, etc. Then, when there was a significant increase in cases from period F to G (Delta Virus period), there was a substantial decrease in the total volume of endoscopy due to the restriction of special endoscopic procedures to only emergency patients, temporary cessation of service operations because many health workers were infected with COVID-19, etc.
Overall, the findings are not surprising, and readers probably want to know whether this decrease is associated with any delay in treatment or disease progression. Answering this question could have significant implications. Can you add more information and analysis results for this part?
Response :
There is no data about the delay of treatment and any disease progression; further analysis can't be performed.
This decrease in the volume of endoscopes impacts decreasing revenues where the endoscopic procedure is one of the non-surgical procedures that are profitable for hospitals. Therefore, the results of this study are expected to be a guideline for preparing SOPs for endoscopic procedures in hospitals during the COVID-19 pandemic so that the total volume of endoscopes can remain stable by the pre-COVID-19 pandemic era. This is explained in the Introduction to the third paragraph, lines 56-63; Discussion section point 4.2 Changes in the Flow of Endoscopic Procedures During the COVID-19 Pandemic (lines 409-422).
Round 2
Reviewer 1 Report
The authors have not improved the significant methodological problems of this paper. My recommendation has to be to reject this manuscript.
Reviewer 2 Report
Thank you for the revision and answering the questions, it is ready to be published.